# Feasibility and Efficacy of Gastric Underwater Endoscopic Mucosal Resection

**DOI:** 10.3390/diagnostics14050536

**Published:** 2024-03-03

**Authors:** Dong Hyun Kim, Seon Young Park, Jin Won Kim, Hyun Soo Kim

**Affiliations:** Department of Internal Medicine, Chonnam National University Hospital and Medical School, Gwangju 61469, Republic of Korea; bono343@cnuh.com (D.H.K.);

**Keywords:** endoscopic mucosal resection, endoscopy, stomach neoplasms, water

## Abstract

Gastric cancer, a leading cause of cancer-related deaths globally, necessitates effective and early detection and treatment strategies. Endoscopic resection techniques, particularly endoscopic mucosal resection (EMR) and endoscopic submucosal dissection (ESD), have evolved significantly, enhancing the treatment of gastric neoplasms. Underwater endoscopic mucosal resection (UEMR) is a widely used technique for the resection of duodenal and colorectal neoplasms. However, the feasibility and efficacy of UEMR in the stomach are not well established. This retrospective observational study, conducted at a tertiary medical center, evaluated the efficacy and safety of UEMR in 81 patients with gastric neoplasms. Thus, it indicates that UEMR is a highly effective and safe technique for managing small to medium-sized gastric neoplasms, achieving 100% en bloc and 93.8% R0 resection rates with a low incidence of complications. Moreover, the procedure time was found to be significantly shorter for UEMR compared to ESD, thus highlighting its efficiency. While UEMR demonstrates high safety and efficacy, it is not suitable for all patients, with some requiring conversion to ESD as a treatment option. Despite the promising results, broader validation through extensive and randomized trials is recommended to establish UEMR as a standard approach in gastric cancer management.

## 1. Introduction

Gastric cancer remains a significant health problem globally, ranking as one of the leading causes of cancer-related deaths, particularly in regions such as East Asia, Eastern Europe, and South America [1,2]. The diverse etiology, encompassing genetic, environmental, and lifestyle factors, contributes to its prevalence and complexity [3,4]. The key to improving survival rates lies in early detection and effective treatment strategies [5,6]. Advances in endoscopic technologies have been pivotal in enhancing early detection, allowing for the identification and treatment of precancerous lesions and early-stage gastric cancer. These technological advancements have revolutionized the approach to treating gastric cancer, by offering more precise diagnostic tools and minimally invasive treatment options, thereby significantly improving patient outcomes and quality of life. The integration of high-definition imaging and artificial intelligence in endoscopy is beginning to offer unprecedented accuracy in diagnosing and managing this disease [7,8].

Endoscopic mucosal resection (EMR) and endoscopic submucosal dissection (ESD) have become established methods in this field [9]. EMR is often the method of choice for treating smaller lesions (less than 10 mm) due to its minimally invasive nature and lower complication rates. However, its effectiveness decreases for larger lesions, leading to reduced rates of en bloc and R0 resections. On the contrary, ESD, despite its higher technical demands and extended procedure duration, allows for the removal of larger, more intricate lesions, ensuring greater rates of complete and margin-free resections. This efficiency positions ESD as the preferred technique for more advanced lesions, although it has major drawbacks such as bleeding and perforation [10,11,12].

Recently, underwater endoscopic mucosal resection (UEMR) has emerged as an innovative technique especially for lesions that were traditionally challenging to be treated with conventional EMR or ESD [13,14,15]. UEMR, characterized by the resection of lesions in a water-filled environment, prevents the need for submucosal injection, thus potentially reducing procedure time and related complications [16]. While UEMR has demonstrated success in treating neoplasms in other organs like the duodenum and large intestine [17,18], its role in gastric neoplasm resection is less explored [19]. Initial studies and case reports have suggested potential advantages of UEMR in gastric lesions [20,21]; however, comprehensive data remain scarce.

In this study, the efficacy, safety, and practical applicability of UEMR in the treatment of gastric neoplasms was explored. By analyzing a series of cases at a tertiary medical center, this study provides valuable insights into the feasibility of UEMR as a standard approach for treating gastric neoplasms, potentially influencing future clinical practice and guidelines in gastric cancer management.

## 2. Materials and Methods

### 2.1. Patient Selection and Data Collection

This retrospective observational study was conducted at a tertiary medical center, focusing on patients who underwent UEMR for treatment of gastric neoplasms between January 2019 and October 2023. Ethical approval was obtained from the Institutional Review Board of Chonnam National University Hospital (IRB number CNUH-2023-372). Due to the retrospective nature of the study, the requirement for informed consent was waived. Patients were included based on the following criteria: patients diagnosed with gastric neoplasms, patients undergoing UEMR during the study period, and patients possessing complete medical records. Data compilation involved extracting patient demographics and medical histories from electronic medical records, specifically, from patients with chronic conditions such as hypertension and diabetes. Medication usage, specifically, documenting the intake of drugs such as aspirin, clopidogrel, and antithrombotics, was carried out carefully. The size of the endoscopic lesions was assessed by comparing the diameter of the snare used for endoscopic resection with the size of the lesions. The shape of the endoscopic lesions was classified according to the Paris classification [22]. The presence of *Helicobacter pylori* (*H. pylori*) was confirmed by polymerase chain reaction testing of the gastric mucosal tissue from two or more sites.

### 2.2. Endoscopic Procedure

The continuation or discontinuation of medications such as aspirin, clopidogrel, and antithrombotics, which patients were already consuming, was determined based on an individualized strategy [23]. This strategy took into account the risk of bleeding associated with the procedure and the risk posed by discontinuing the medication. All UEMR procedures were conducted by a single experienced endoscopist (Kim DH) using standardized equipment and techniques. The equipment included a high-definition RGB sequential video-endoscopy system (EVIS LUCERA ELITE; Olympus, Tokyo, Japan) and cap-assisted esophagogastroduodenoscopy (GIF-HQ290, Olympus, Tokyo, Japan). Narrow-band imaging (NBI) was employed to delineate the margins of the neoplasms. Sedation protocol involved initial administration of midazolam (2–3 mg) and pethidine (25 mg), followed by propofol as needed for pain control. Oxygen was supplied at 2 L/min, and patients’ vital signs were continuously monitored. During the procedure, CO_2_ insufflation was routinely used.

UEMR was performed without submucosal injection instead of following conventional EMR. The UEMR technique involved several steps, such as: (1) marginal delineation using narrow-band imaging (NBI); (2) lesion submersion using a water-jet pump; and (3) lesion snaring and resection with snare and electrocautery (VAIO 3, ERBE Co. Ltd., Tubingen, Germany). The snare used was either a 15 mm diameter hexagonal snare (SnareMaster Plus, model SD-400U-15, Olympus, Tokyo, Japan) or a 20 mm diameter hexagonal snare (Endo-Upex Electrosurgical Snare, model RSH-2320 (SC), UpexMed, Gyeonggi-do, Republic of Korea). The settings for the VAIO 3 were configured as follows: Endocut Q mode, effect level 2, an incision duration of 3. In cases of immediate bleeding (Figure 1), hemostasis was achieved using either snare tip or electrosurgical hemostatic forceps (Coagrasper^®^ FD-411UR, Olympus, Tokyo, Japan). An endoscopic clip was also used in cases where hemostasis was not achieved by the usual methods. During gastric UEMR, saline was used for water injection, with the volume of injected saline ranging between 100 to 400 mL. After the procedure was completed, the remaining water in the gastric lumen was suctioned out using endoscopy. Failure cases, necessitating a switch to ESD, were identified as those where snaring the lesion was unsuccessful after two attempts (Figure 2) (Appendix A {This video showcases an example of gastric underwater endoscopic mucosal resection performed on a 10-mm flat lesion with a central depression, located on the lesser curvature side of the antrum}).

Resected specimens were preserved in 10% formalin and subjected to histopathological examination. Parameters such as histological type, depth of invasion, and margin involvement were assessed. R0 resection was characterized by en bloc resection with histologically clear margins. Procedure time was defined as the time from the use of sedative drugs to the completion of endoscopic resection, bleeding control, suction of fluid in gastric lumen, and extubation of the endoscope.

### 2.3. Adverse Events and Follow-Up Strategy

During the procedure, hypotension was identified when systolic blood pressure dropped below 90 mmHg. A decline in arterial oxygen saturation below 90% for at least 10 s was classified as hypoxia. Immediate bleeding referred to active hemorrhage observed during the immediate post-resection phase. However, post-procedure, delayed bleeding was recognized if it required a blood transfusion, emergency endoscopy, or resulted in a hemoglobin drop of over 2 g/dL [24]. A perforation related to the procedure was confirmed through direct endoscopic visualization of an extraluminal space or the detection of free air in the abdominal cavity via radiography or CT scans [25]. Post-EMR coagulation syndrome (PECS) was characterized by signs of inflammation such as abdominal discomfort linked to the procedure, fever (≥37.6 °C), white blood cell count of ≥10,000 cells/μL, or elevated C-reactive protein levels (≥0.5 mg/dL), in the absence of any signs of post-EMR perforation [26]. Atelectasis was identified by comparing post-procedure chest radiograms with those taken before the endoscopic resection, irrespective of the presence of clinical symptoms. Radiological indicators of atelectasis included both direct and indirect signs. Direct signs included the convergence of pulmonary vessels, clustered air bronchograms, and shifting of the interlobar fissures. Indirect signs were opacification of the lung tissue and upward movement of the diaphragm on the affected side [27]. Routine follow-up endoscopies were planned for approximately 3 to 6 months after resection to monitor for any recurrence, with a thorough examination and biopsy of the resected area.

### 2.4. Statistical Analysis

In this study, descriptive statistics were used to summarize patient demographics and lesion characteristics in both the gastric UEMR group and the ESD conversion group. This included details such as comorbidities, medication history, lesion size, and morphology and outcomes from the endoscopic resection. The analysis quantified frequencies of complete (en bloc) and margin-free (R0) resections along with procedural and respiratory complications and recurrence rates. Continuous variables were analyzed using independent *t*-tests, while chi-square or Fisher’s exact tests were applied to categorical data. A *p*-value less than 0.05 was considered indicative of statistical significance. Statistical evaluations were performed using IBM SPSS software, version 25 (IBM Corp., Armonk, NY, USA).

## 3. Results

### 3.1. Baseline Characteristics of Patients and Gastric Tumors

A total of 81 patients underwent gastric UEMR. The mean age of the patients was approximately 64.9 ± 8.3 y, with male predominance (51 males, 30 females). Moreover, 6.2% of patients were taking aspirin, 7.4% were taking antiplatelet therapy, and 2.5% were taking anticoagulants. The most common comorbidity observed was hypertension (40.7%) followed by diabetes mellitus (23.5%). A history of *H. pylori* infection was present in 58.0% of the cases.

With respect to gastric tumors, the mean lesion size was 10.2 ± 2.9 mm (range 6–18 mm). The antrum was the most common tumor location (63.0%) with the lesser curvature being the most frequent circumferential location (40.7%) followed by the corpus (18.5%). The predominant tumor morphology was IIa (42.0%) followed by IIc (30.9%). Most procedures were performed under sedative endoscopy using midazolam (92.6%) and propofol (96.3%) (Table 1).

### 3.2. Treatment Outcomes

All 81 patients successfully underwent endoscopic resection. Of these, 76 were treated with UEMR and 5 (6.2%) required conversion to ESD. The mean procedure time for UEMR was significantly shorter (9.0 ± 3.6 min) compared to the ESD conversion cases (25.7 ± 6.5 min, *p* < 0.01). The dosage of midazolam did not differ significantly between the UEMR and ESD conversion groups (2.8 ± 0.5 mg vs. 3.0 ± 0 mg, *p* = 0.45); however, the use of propofol was significantly higher in the ESD conversion group (166.0 ± 72.0 mg) compared to the UEMR group (43.6 ± 26.8 mg, *p* < 0.01). En bloc resection was achieved in all cases with a 93.8% R0 resection rate across the entire cohort (UEMR 93.4% vs. ESD conversion 100%, *p* = 0.84). Histologically, most lesions were characterized as low-grade dysplasia (87.7%), followed by high-grade dysplasia (8.6%), and adenocarcinoma (3.7%). Snare tips (46.9%) were most commonly used to control immediate bleeding, followed by hemostatic forceps (8.6%) and endoscopic clips (4.9%). Although not statistically significant, hemostasis using a snare tip was used more often in the UEMR group (48.7%) than in the ESD conversion group (20.0%, *p* = 0.36), and, conversely, hemostatic forceps were used more often in the ESD conversion group (40.0%) than in the UEMR group (6.6%). Significantly, more endoscopic clips were performed in the ESD conversion group (40%) than in the UEMR group (2.6%, *p* = 0.02).

Three cases of adenocarcinoma were reported. The pathological findings in all cases showed well-differentiated adenocarcinoma invading the lamina propria (stage pT1a), with clear lateral and vertical margins, indicating no residual tumor cells. In addition, there were no signs of lymphovascular or perineural invasion in any case. Furthermore, none of the patients experienced adenocarcinoma recurrence. Two patients underwent Rx resection at the lateral margins, which indicated that the lesion margins could not be assessed. Both patients were histologically diagnosed with low-grade dysplasia. Three patients underwent R1 resection; among them, one had low-grade dysplasia and the remaining two had high-grade dysplasia. In all three cases, tumor cells were present at the lateral margin, but absent at the vertical margin. Recurrence occurred in two patients, both of whom had undergone R1 resection with positive findings for tumor cells at the lateral margin. The pathological reports of these cases confirmed high-grade dysplasia, and both patients subsequently underwent a second endoscopic resection (ESD). None of the patients experienced disease recurrence after this secondary intervention (Table 2).

### 3.3. Adverse Events

Immediate bleeding occurred in 37.0% of the cases; however, it was effectively managed without any reported cases of delayed bleeding. There was no significant difference in the occurrence of immediate bleeding between the UEMR group (36.8%) and the ESD conversion group (40%, *p* = 0.62). Atelectasis was observed in 7.4% of patients, slightly more frequent in the ESD conversion group (20.0%) compared to the UEMR group (6.6%, *p* = 0.33). There were no reported cases of hypoxemia, perforation, or post coagulation syndrome during the procedure (Table 3).

### 3.4. Follow-Up

During the follow-up period, approximately 63.0% of patients were monitored. Recurrence was observed in 2 patients (2.6%), for whom additional secondary endoscopic resection through ESD was performed. The two patients with confirmed recurrence were those who underwent UEMR, and no recurrence was confirmed in patients who underwent conversion to ESD. Two patients in the UEMR arm relapsed and underwent secondary ESD at a later date.

## 4. Discussion

UEMR is a distinct approach compared to conventional EMR. In UEMR, instead of using submucosal injection, the procedure involves suctioning out air and filling the area around the lesion with water. This creates buoyancy, causing the lesion to naturally float upwards. The lesion is then safely captured with a snare and removed using electrocautery. This technique leverages the buoyant properties of the lesion in the fluid-filled environment, which facilitates easier and potentially safer snaring and resection of the lesion [16]. Recently, the use of UEMR has significantly increased for the treatment of conditions such as colorectal neoplasms and superficial non-ampullary duodenal neoplasms [15,17]. On the contrary, in the case of gastric lesions, there have been only case reports and studies with a small number of subjects [19,20,21]. This has limited analysis of the feasibility and effectiveness of UEMR in the treatment of gastric neoplasms.

This study represents the largest research conducted on patients undergoing gastric UEMR compared to previous studies in this field. Additionally, it holds significance in analyzing both the successful and failed cases of UEMR, thereby identifying patient groups for whom UEMR was effective and those groups for whom it was not. UEMR was performed on 81 patients with gastric neoplasms ranging in size from 6 to 18 mm. Out of these patients, UEMR was successfully performed in 76 individuals (93.8%), while in 5 patients (6.2%), the lesion could not be captured with a snare in the underwater state. In 6.2% of cases, snaring the lesion effectively underwater proved challenging, leading to a switch to ESD. This finding underscores the feasibility of gastric UEMR, demonstrating its effectiveness in a majority of cases while also highlighting the limitations in certain situations.

In patients who underwent UEMR, the mean size of the lesions was 10.1 mm, while for those who required a conversion to ESD, the mean lesion size was found to be slightly larger, i.e., 11.6 mm. However, this difference was not statistically significant. Similarly, no statistical significance was found in terms of the location of the lesions. Moreover, it was noted that in all 5 cases requiring ESD conversion, the lesions were located in the gastric antrum. In contrast, for successful UEMR cases, 60.5% of the lesions were in the antrum and 19.7% in the corpus. The reasons for the occurrence of ESD conversion cases in the antrum are not entirely clear. However, during the actual procedure, filling the area around the lesion with water and the subsequent expansion of the stomach wall can make it difficult for the snare to approach closely, thus complicating UEMR. In actual ESD conversion cases, the stomach often exhibited conditions like a cascade stomach, where the stomach expands well. Such anatomical characteristics of the stomach might have contributed to the difficulty in performing gastric UEMR effectively.

In general, gastric EMR shows varying outcomes based on the size of the lesion. For lesions smaller than 10 mm, en bloc resection rates are reported at around 91%, with complete resection rates at approximately 80%. However, for lesions larger than 10 mm, the R0 resection rate drops to about 64%, and the en bloc resection rate decreases to roughly 51.5% [28]. In results reported for modified EMR with precutting, for lesions smaller than 10 mm, en bloc resection and R0 resection rates are both reported to be around 87%. In the case of lesions between 10–19 mm, en bloc resection is reported at 83%, and R0 resection at about 81%. This data suggests that the size of the gastric lesion significantly impacts the success rates of EMR procedures, including those with modifications such as precutting [29]. In this study, among the 76 patients for whom gastric UEMR was feasible, 100% en bloc resection was achieved, and R0 resection was successful in 93.4% of the cases. Considering that ESD conversion was performed in 5 patients (6.2%), and the size of the lesions ranged from 6 to 18 mm, the therapeutic outcomes of gastric UEMR seem to be comparable or slightly better than those of conventional EMR or EMR with precutting. This suggests that UEMR might be a viable alternative with potentially better or, at least, similar efficacy in treating certain gastric lesions, particularly in the specified size range. As the visibility of the mucosal pattern is more magnified in the underwater state, the mucosal pattern is more visible, and the neoplastic and non-neoplastic parts are more clearly distinguished, which is thought to be one of the reasons for the high en bloc and R0 resection results. In the two cases of recurrence, both patients underwent R1 resection, where tumor cells were detected at the lateral margin. Pathology reports highlighted high-grade dysplasia along with the presence of residual tumor cells at the lateral margin. An inability to achieve R0 resection is associated with recurrence. The procedural image was reviewed; however, there were no visual defects during the procedure, and there were no residual lesions in the post-procedure image. It may have been difficult to assess the presence or absence of residual lesions using endoscopic images because of the cautery effect.

Moreover, in terms of endoscopic resection time, UEMR demonstrated a significantly shorter procedure duration. While conventional EMR showed a procedure time of 13.9 min for lesions smaller than 10 mm and 25.8 min for lesions between 10 and 20 mm [28], UEMR had a much shorter average procedure time of only 9.0 min. This is comparatively shorter than ESD, which takes about 42.4 min for lesions smaller than 10 mm and up to 84 min for lesions between 10 and 20 mm. Thus, it highlights the efficiency of UEMR as a technique, offering a quicker alternative to both conventional EMR and ESD [28]. UEMR likely has a shorter procedure time compared to EMR or ESD as it eliminates steps like the insertion of an injector for submucosal injection. Instead, UEMR directly employs water infusion to fill the area and enables snaring, streamlining the overall process. During gastric endoscopic resection, shorter procedure times are associated with fewer respiratory complications [30]. This is particularly important for elderly patients and those with underlying health conditions, who are more prone to respiratory complications [27]. Therefore, in such cases, the quick procedure time offered by UEMR could be a significant advantage.

With respect to adverse events, about 36.8% of the patients who underwent UEMR experienced immediate bleeding; however, endoscopic hemostasis was successful in all cases, and there were no instances of delayed bleeding. There were also no occurrences of perforation, hypotension, hypoxia, or post-coagulation syndrome. However, asymptomatic atelectasis occurred in approximately 6.6% of cases, a complication that can also arise after conventional gastric EMR or ESD. Specifically, in UEMR, normal saline is used for fluid injection into the gastric lumen, and all procedures were performed in the left lateral decubitus position. This led to the development of atelectasis in the left lower lung field in some patients. However, these instances were temporary and occurred without fever or respiratory symptoms, indicating that while atelectasis is a potential complication of UEMR, it is generally manageable and transient. Compared to previous studies where gastric EMR and ESD have shown a perforation risk between 0–5% and a risk of massive bleeding in the range of 3–5% [28,29], UEMR demonstrates relatively higher safety. The absence or markedly lower incidence of such serious complications in UEMR suggests its advantage in terms of safety.

This study was significant in evaluating the feasibility, efficacy, and safety of gastric UEMR for small to intermediate-sized lesions. As 81 cases were studied, it involved a larger patient cohort compared to previous studies, marking progress in analyzing treatment effectiveness. However, as a retrospective study conducted by a single endoscopist, it may be subject to biases related to the operator and the procedural environment. Additionally, being a non-randomized controlled trial, it lacked the ability to directly compare with conventional EMR, representing a limitation in drawing comprehensive comparative conclusions.

This study demonstrated that gastric UEMR has high en bloc and R0 resection rates, with a notably lower incidence of adverse events. The significantly reduced procedure time would be particularly beneficial for elderly patients or those with multiple underlying health conditions. However, many patients were found to have low grade dysplasia rather than advanced lesions. In three cases of adenocarcinoma, all demonstrated R0 resection and no recurrence was observed. However, the depth of invasion was limited to the lamina propria, indicating a less invasive cancer, and the number of cases was small. If the UEMR was performed in cases with early gastric cancer, the complete resection rate may decrease and the incidence of recurrence may increase, even for small lesions. Moreover, endoscopic follow-up was not performed for all patients; therefore, the recurrence rate may have been higher than that currently identified. It is important to recognize that UEMR may not be suitable for all patients, and, in some instances, a switch to ESD might be required. Further research including randomized controlled trials comparing UEMR with conventional EMR is necessary to more comprehensively evaluate the efficacy and safety of gastric UEMR.

## 5. Conclusions

UEMR has demonstrated effectiveness and safety for treating small to medium-sized gastric neoplasms, with favorable results in resection rates and complication management. However, to validate the role of UEMR in broader clinical settings and to establish it as a standard treatment option, further comprehensive, randomized trials are essential. This will ensure a more thorough assessment of its efficacy and safety across diverse patient populations.

## Figures and Tables

**Figure 1 diagnostics-14-00536-f001:**
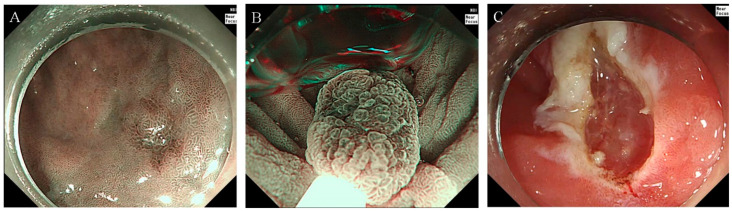
Gastric UEMR procedure example (**A**) narrow-band imaging reveals an 8-mm flat lesion with a central depression located at the great curvature of the antrum. (**B**) The lesion being snared in an underwater state during the UEMR procedure. (**C**) Post-UEMR view shows the site after successful removal of the neoplasm. Abbreviation: UEMR, underwater endoscopic mucosal resection.

**Figure 2 diagnostics-14-00536-f002:**
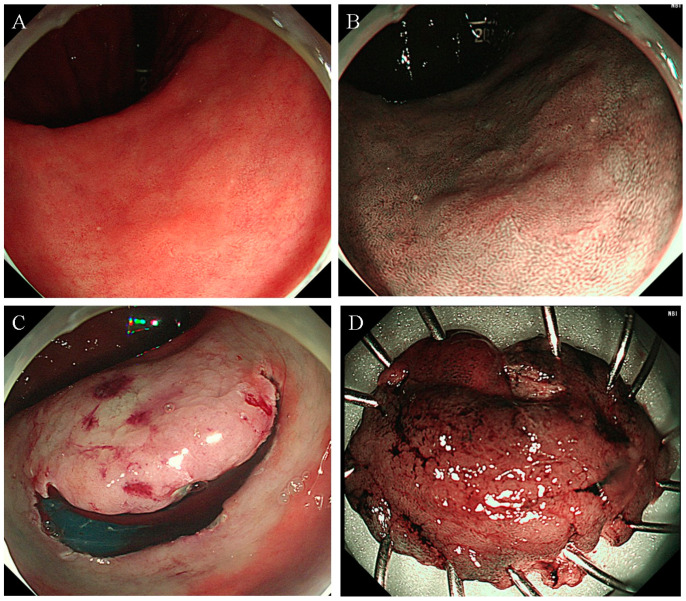
Failed UEMR case leading to conversion to ESD. (**A**,**B**) An approximately 18-mm flat neoplasm situated at the lesser curvature of the antrum, illustrating the technical challenge of UEMR. (**C**) The neoplasm undergoing ESD following an unsuccessful UEMR attempt. (**D**) The resected specimen post-ESD procedure. Abbreviation: ESD, endoscopic submucosal dissection; UEMR, underwater endoscopic mucosal resection.

**Table 1 diagnostics-14-00536-t001:** Baseline profile of gastric neoplasms undergoing UEMR.

	All Patients (*n* = 81)	UEMR (*n* = 76)	ESD Conversion (*n* = 5)	*p*-Value
Age, years	64.9 ± 8.3	65.3 ± 8.4	60.2 ± 5.3	0.10
Female	30 (37.0)	30 (39.5)	0 (0)	0.15
Tumor size, mm	10.2 ± 2.9	10.1 ± 2.8	11.6 ± 4.1	0.45
Tumor location				0.37
Cardia	4 (4.9)	4 (5.3)	0	
Corpus	15 (18.5)	15 (19.7)	0	
Antrum	51 (63.0)	46 (60.5)	5 (100)	
Pylorus	11 (13.6)	11 (14.5)	0	
Morphology				0.25
Is	12 (14.8)	12 (15.8)	0	
IIa	34 (42.0)	33 (43.4)	1 (20.0)	
IIb	8 (9.9)	6 (7.9)	2 (40.0)	
IIc	25 (30.9)	23 (30.3)	2 (40.0)	
III	2 (2.5)	2 (2.6)	0	
Comorbidity				
Hypertension	33 (40.7)	29 (38.2)	4 (80.0)	0.05
Diabetes	19 (23.5)	19 (25.0)	0	0.57
*H pylori* infection	47 (58.0)	42 (55.3)	5 (100)	0.28
Medication				
Aspirin	5 (6.2)	5 (6.6)	0	0.56
Clopidogrel	6 (7.4)	5 (6.6)	1 (20.0)	0.33
Antithrombotics	2 (2.5)	2 (2.6)	0	0.61
Sedation				
Midazolam	75 (92.6)	70 (92.1)	5 (100)	0.51
Propofol	78 (96.3)	73 (96.1)	5 (100)	0.65

Data are shown as mean ± standard deviation or *n* (%); ESD, endoscopic submucosal dissection; *H pylori*, *Helicobacter pylori*; UEMR, underwater endoscopic mucosal resection.

**Table 2 diagnostics-14-00536-t002:** Treatment outcomes in gastric UEMR and ESD conversion procedures.

	All Patients (*n* = 81)	UEMR (*n* = 76)	ESD Conversion (*n* = 5)	*p*-Value
Procedure time (min)	9.9 ± 5.3	9.0 ± 3.6	25.7 ± 6.5	<0.01
Midazolam (mg)	2.8 ± 0.5	2.8 ± 0.5	3.0 ± 0	0.45
Propofol (mg)	51.1 ± 42.6	43.6 ± 26.8	166.0 ± 72.0	<0.01
En bloc resection	81 (100)	76 (100)	5 (100)	1.0
R0 resection	76 (93.8)	71 (93.4)	5 (100)	0.84
Histology				0.69
LGD	71 (87.7)	66 (86.8)	5 (100)	
HGD	7 (8.6)	7 (9.2)	0	
Adenocarcinoma	3 (3.7)	3 (3.9)	0	
Hemostatic device use				
Snare tip	38 (46.9)	37 (48.7)	1 (20.0)	0.36
Hemostatic forceps	7 (8.6)	5 (6.6)	2 (40.0)	0.06
Endoscopic clip	4 (4.9)	2 (2.6)	2 (40.0)	0.02
Follow-up				
Recurrence	2 (2.5)	2 (2.6)	0	0.88
Secondary endoscopic resection	2 (2.5)	2 (2.6)	0	0.94

Data are shown as mean ± standard deviation or *n* (%); ESD, endoscopic submucosal dissection; HGD, high grade dysplasia; LGD, low grade dysplasia; UEMR, underwater endoscopic mucosal resection.

**Table 3 diagnostics-14-00536-t003:** Adverse events in gastric UEMR and ESD conversion procedures.

	All Patients (*n* = 81)	UEMR (*n* = 76)	ESD Conversion (*n* = 5)	*p*-Value
Immediate bleeding	30 (37.0)	28 (36.8)	2 (40.0)	0.62
Delayed bleeding	0	0	0	
Hypotension	0	0	0	
Hypoxemia	0	0	0	
Atelectasis	6 (7.4)	5 (6.6)	1 (20.0)	0.33
Perforation	0	0	0	
PECS	0	0	0	

Data are shown as mean ± standard deviation or *n* (%); EMR, endoscopic mucosal resection; ESD, endoscopic submucosal dissection; HGD, high grade dysplasia; LGD, low grade dysplasia; PECS, post-EMR coagulation syndrome; UEMR, underwater endoscopic mucosal resection.

## Data Availability

The data are not publicly available due to privacy and ethical restrictions. Data presented in this study are available upon request from the corresponding author.

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
