# Peer review of "Feasibility and Efficacy of Gastric Underwater Endoscopic Mucosal Resection"

_diagnostics, 2024, doi:10.3390/diagnostics14050536_

Round 1

Reviewer 1 Report

Comments and Suggestions for Authors

Thank you for the opportunity to review this article. The authors investigated the feasibility of UEMR for gastric tumors in 81 patients. They demonstrated that UEMR can be performed safely and the R0 resection rate was as high as 93.4%, thus demonstrating the feasibility of UEMR. However, there are significant concerns about acceptance, which are described below.

1. As the authors note in the discussion, only 63% of the cases were followed up, which is insufficient to describe short-term outcomes. In other words, the focus is only on the experience of being able to perform UEMR, which is hardly a novel finding.

2. Do you use only demarcation diagnosis based on NBI observation, so that marking is not necessary? Does this mean that the lesions that underwent UEMR were predominantly elevated lesions with well-defined tumor margins? 

3. Please describe the details of the snare used.

4. There is too little histopathologic information to evaluate this technique. What was the reason for not achieving R0; was R1 associated with lateral or vertical margins?

5. How would you consider the reason for the recurrence in two of the cases? Please discuss this point.

6. One of the most important things for the endoscopist is to obtain a specimen that allows assessment of tumor depth. How far from the muscularis mucosae can be histopathologically resected with UEMR?  If there is unexpected submucosal invasion, it cannot be accurately evaluated, and there is a risk of local recurrence or distant metastasis.

Author Response

Reviewer 1.

Thank you for the opportunity to review this article. The authors investigated the feasibility of UEMR for gastric tumors in 81 patients. They demonstrated that UEMR can be performed safely and the R0 resection rate was as high as 93.4%, thus demonstrating the feasibility of UEMR. However, there are significant concerns about acceptance, which are described below.

  1. As the authors note in the discussion, only 63% of the cases were followed up, which is insufficient to describe short-term outcomes. In other words, the focus is only on the experience of being able to perform UEMR, which is hardly a novel finding.

=> Thank you for your comments. Not all patients were followed-up endoscopically. Approximately 63% of patients had a follow-up endoscopy, and 37% did not. This may have led to underestimation of the recurrence rate.

We have commented on this limitation in the Discussion section (page 10, line 359–360) as "Moreover, endoscopic follow-up was not performed for all patients; therefore, the recurrence rate may have been higher than that currently identified. "

  1. Do you use only demarcation diagnosis based on NBI observation, so that marking is not necessary? Does this mean that the lesions that underwent UEMR were predominantly elevated lesions with well-defined tumor margins? 

=> Thank you for your helpful comment. We examined the lesion in NBI mode before underwater EMR without marking. When we turned on the NBI mode underwater, the lesion appeared magnified, similar to a magnifying glass, and the pattern of the mucosa was more clearly distinguishable between the neoplastic and non-neoplastic areas. Figures 1a and 1b show the difference in the visibility of the mucosal pattern between the air insufflation and underwater conditions.

Therefore, underwater EMR was performed without special markings. When performing conventional EMR (injection and snaring with electrocautery) rather than ESD, we also performed it without marking before endoscopic resection.

We added the comments in Discussion section (page 9, line 301–304) as “As the visibility of the mucosal pattern is more magnified in the underwater state, the mucosal pattern is more visible, and the neoplastic and non-neoplastic parts are more clearly distinguished, which is thought to be one of the reasons for the high en bloc and R0 resection results.”

  1. Please describe the details of the snare used.

=> Thank you for the thoughtful comment. We used two types of snare (15 mm sized, Hexagonal snare, SnareMaster Plus, SD-400U-15, Olympus®, Tokyo, Japan or 20 mm sized, Hexagonal snare, Endo-Upex Electrosurgical Snare, RSH-2320(SC), UpexMed® Gyeonggi-do, Republic of Korea).

The following are described in the METHODS section at line 99–102, page 3:

“The snare used was either a 15 mm diameter Hexagonal snare (SnareMaster Plus, model SD-400U-15, Olympus, Tokyo, Japan) or a 20 mm diameter Hexagonal snare (Endo-Upex Electrosurgical Snare, model RSH-2320 (SC), UpexMed, Gyeonggi-do, Republic of Korea).”

  1. There is too little histopathologic information to evaluate this technique. What was the reason for not achieving R0; was R1 associated with lateral or vertical margins?

Thank you for your feedback. Two patients underwent Rx resection on the lateral margin, which made it impossible to evaluate the margin of the lesions. The histological findings in both cases were low-grade dysplasia. We reviewed the images of the procedure and observed a cautery effect at the cutting edge when using a snare. However, there was no recurrence in patients who underwent Rx resection.

The remaining three patients underwent R1 resection and all had gastric adenoma. Two patients had Paris classification IIa morphological adenoma and one patient had IIc morphological adenoma. The tumor sizes were 16 mm, 16 mm, and 12 mm. All three adenomas (two high-grade dysplasia and one low-grade dysplasia) were positive for tumor cells at the lateral margin and negative at the vertical margin. We reviewed the procedure images; however, there were no visual defects during the procedure, and there were no residual lesions in the post-procedure WLI and NBI mode images. It may have been difficult to assess the presence or absence of residual lesions using WLI and NBI mode images owing to the cautery effect. We believe that this is a limitation of UEMR.

These aspects were described in the Results and Discussion sections as follows:

“Two patients underwent Rx resection at the lateral margins, which indicated that the lesion margins could not be assessed. Both patients were histologically diagnosed with low-grade dysplasia. Three patients underwent R1 resection; among them, one had low-grade dysplasia and the remaining two had high-grade dysplasia. In all three cases, tumor cells were present at the lateral margin, but absent at the vertical margin. Recurrence occurred in two patients, both of whom had undergone R1 resection with positive findings for tumor cells at the lateral margin. The pathological reports of these cases confirmed high-grade dysplasia, and both patients subsequently underwent a second endoscopic resection (ESD). None of the patients experienced disease recurrence after this secondary intervention (Results section, line 210–219, Page 6-7)

“The procedural image was reviewed; however, there were no visual defects during the procedure, and there were no residual lesions in the post-procedure image. It may have been difficult to assess the presence or absence of residual lesions using endoscopic images because of the cautery effect.” (Discussion, line 308–311, page 9)

  1. How would you consider the reason for the recurrence in two of the cases? Please discuss this point.

=> In the two cases of recurrence, both had R1 resection with positive tumor cells found at the lateral margin. Both cases had pathological reports indicating high-grade dysplasia and the presence of remnant tumor cells at the lateral margin, which is a major factor in recurrence. Gastric ESD was performed for the recurrent lesions, resulting in successful R0 resection. No patient experienced recurrence after the ESD procedure.

These aspects were described in the Results section and Discussion section as follows:

“Recurrence occurred in two patients, both of whom had undergone R1 resection with positive findings for tumor cells at the lateral margin. The pathological reports of these cases confirmed high-grade dysplasia, and both patients subsequently underwent a second endoscopic resection (ESD). None of the patients experienced disease recurrence after this secondary intervention” (Results section, line 215–219, page 6-7)

“In the two cases of recurrence, both patients underwent R1 resection, where tumor cells were detected at the lateral margin. Pathology reports highlighted high-grade dysplasia along with the presence of residual tumor cells at the lateral margin. An inability to achieve R0 resection is associated with recurrence.” (Discussion section, line 304-308, page 9)

  1. One of the most important things for the endoscopist is to obtain a specimen that allows assessment of tumor depth. How far from the muscularis mucosae can be histopathologically resected with UEMR?  If there is unexpected submucosal invasion, it cannot be accurately evaluated, and there is a risk of local recurrence or distant metastasis.

=> Thank you for your comments.

There were three cases of adenocarcinoma. All pathological reports indicated a well-differentiated adenocarcinoma with lamina propria invasion (pT1a) and no residual tumor cells at the lateral and vertical margins. There was no evidence of lymphovascular or perineural invasion. None of the patients experienced adenocarcinoma recurrence.

The description included in the Results section is as follows:

“Three cases of adenocarcinoma were reported. The pathological findings in all cases showed well-differentiated adenocarcinoma invading the lamina propria (stage pT1a), with clear lateral and vertical margins, indicating no residual tumor cells. In addition, there were no signs of lymphovascular or perineural invasion in any case. Furthermore, none of the patients experienced adenocarcinoma recurrence.” (Results section, Line 206–210, Page 6).

Reviewer 2 Report

Comments and Suggestions for Authors

only 3 of 76 UEMR cases had a histologic diagnosis of adenocarcinoma. The histologic subtype also needs to be stated, and also clarify how these 3 cases correlated with the endoscopic classification as well as the margin status. It is important to analyze if this procedure is adequate treatment for gastric adenocarcinomas.

Author Response

Reviewer 2.

Comments and Suggestions for Authors

only 3 of 76 UEMR cases had a histologic diagnosis of adenocarcinoma. The histologic subtype also needs to be stated, and also clarify how these 3 cases correlated with the endoscopic classification as well as the margin status. It is important to analyze if this procedure is adequate treatment for gastric adenocarcinomas.

=> Thank you for your comments.

Two of the three patients had Paris classification IIa morphology, and the remaining patients had IIc morphology. The tumor sizes were 11 mm, 11 mm, and 10 mm. No procedural disruptions, including visual disturbances or fibrosis, were observed. We achieved en bloc and R0 resections in all the adenocarcinoma cases. All pathological reports indicated a well-differentiated adenocarcinoma with lamina propria invasion (pT1a) and no residual tumor cells at the lateral and vertical margins. There was no evidence of lymphovascular or perineural invasion. None of the patients experienced recurrence of adenocarcinoma. Further details are provided in the Results section.

“Three cases of adenocarcinoma were reported. The pathological findings in all cases showed well-differentiated adenocarcinoma invading the lamina propria (stage pT1a), with clear lateral and vertical margins, indicating no residual tumor cells. In addition, there were no signs of lymphovascular or perineural invasion in any case. Furthermore, none of the patients experienced adenocarcinoma recurrence.” (Results section, Line 206–210, Page 6).

“In three cases of adenocarcinoma, all demonstrated R0 resection and no recurrence was observed. However, the depth of invasion was limited to the lamina propria, indicating a less invasive cancer, and the number of cases was small.” Discussion section, line 354–357, page 10)

Round 2

Reviewer 1 Report

Comments and Suggestions for Authors

The manuscript has been revised well. I think the manuscript will be acceptable.